# Key Factors in the Implementation of Wearable Antennas for WBNs and ISM Applications: A Review WBNs and ISM Applications: A Review

**Fatimah Fawzi Hashim** [1,2]**, Wan Nor Liza Binti Mahadi** [1,2,*]**, Tariq Bin Abdul Latef** [1,2] **and Mohamadariff Bin Othman** [1,2]

[1] Department of Electrical Engineering, University Malaya, Kuala Lumpur 50603, Malaysia
[2] Electromagnetic Radiation & Devices Research (EMRD), Faculty of Engineering, University Malaya, Kuala Lumpur 50603, Malaysia
[*] Correspondence: wnliza@um.edu.my

**Abstract:** The increasing usage of wireless technology has prompted the development of a new generation antenna compatible with the latest devices, with on-body antennas (wearable antennas) being one of the revolutionary applications. This modern design is relevant in technologies that require close human body contact, such as telemedicine and identification systems, due to its superior performance compared to normal antennas. Some of its finer characteristics include flexibility, reflection coefficient, bandwidth, directivity, gain, radiation, specific absorption rate (SAR), and efficiency that are anticipated to be influenced by the coupling and absorption by the human body tissues. Furthermore, improvements like band-gap structure and artificial magnetic conductors (AMC) and (DGS) are included in the wearable antenna that offers a high degree of isolation from the human body and significantly reduces SAR. In this paper, the development of on-body antennas and how they are affected by the human body were reviewed. Additionally, parameters that affect the performance of this new antenna model, such as materials and common technologies, are included as an auxiliary study for researchers to determine the factors affecting the performance of the wearable antenna and the access to a highly efficient antenna.

**Keywords:** AMC; cloth substrate; DGS; EBG; PBG; SAR; wearable antenna

## 1. Introduction

The wearable/on-body antenna has undergone rapid development over the years because of its essential application, especially in the medical field, where these devices are directly connected to the human body with remote control features. Figure 1 shows the wireless body area [1]. Antennas are the most important part of these devices, where signals are exchanged without damaging the human body and its environment. Its main function is to track an individual's health status while being worn on various parts of the body. The new generation of antennas has several advantages, including lightweight, twistable, flexibility and steady performance even when fixed on the irregularly shaped human body.

The human body experiences constant changes in terms of shape and conductivity. When the magnetic field enters and is distributed throughout the body, it is influenced by the body's physiology, frequency, and polarization, thus enabling the antenna to continuously detect the body compared to its environment [2]. The antenna gain and efficiency are affected by the absorption of the human body because the gain affects the energy sent with radiation direction. Therefore, researchers are attempting to develop various techniques to achieve a high antenna gain leading to maximum efficiency.

The substrate dielectric constant and its thickness decide the bandwidth and efficiency of a patch antenna. The dielectric constant of the substrate should range between

$2.2 \leq \varepsilon r \leq 12$, which lowers the dielectric constant and increases spatial waves; the decreasing of the surface wave losses is contributed by the low dielectric constant connected to guided wave propagation within the antenna substrates and that resulting in a high impedance bandwidth of the antenna [2]. Meanwhile, substrate thickness normally ranges from 0.003 λ to 0.005 λ, where λ is the wavelength used to increase antenna bandwidth [3]. Regarding the dissipation factor (Loss tangent tan δ), the higher the loss tangent values, the more lossy the dielectric substrate will be, thus determining radiation efficiency [2].

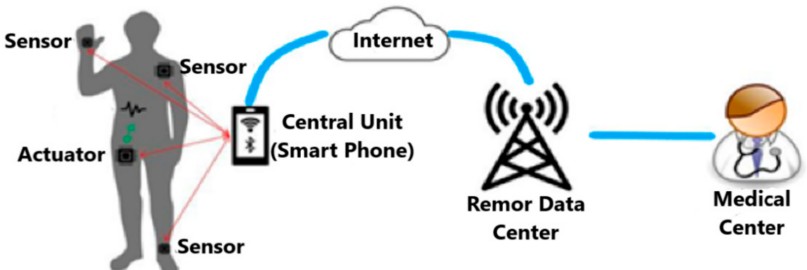

**Figure 1.** Wireless Body Area, adapted from [1].

In this paper, the wearable antenna history and specific absorption rate (SAR), antenna radiation on the human body, materials and methods used in their manufacture are discussed. Then, the effect of these factors on the antenna gain and the common techniques and styles used to obtain a high gain suitable for applications related to the human body were analysed.

## 2. Historical Review of Wearable/Cloth Antenna

Wearable antennas are not new to the market, but their lack of efficiency has limited their application in performing the increasingly complex tasks required over the years. Researchers and manufacturers continue to search for new methods to enhance its features for better utilisation. Nevertheless, several challenges needed to be overcome, including curves of the human body and environmental factors such as humid weather that may directly affect the device's performance. Measures that have been taken to improve wearable antennas' performance include utilising cloth material, installing reflectors adjacent to the antenna and position slots in the antenna ground. The focus is now on designing antennas that can be implanted in clothing such as hats, shirts or shoes to encourage human-related applications.

*Dielectric Permittivity Parameter Measurement*

Textile dielectric permittivity can be measured in numerous ways, and each can be determined using different algorithms. The correct choice affects accuracy, convenience, frequency range, measuring speed, etc. [4].

Methods for measuring permittivity:

- Parallel plate method;
- Transmission/reflection line method;
- Open-ended probe method;
- Free space method;
- Resonant (Cavity) Technique;
- Microstrip patch antenna covered with the material under test.

High-frequency permeability is measured using the resonant (cavity) approach; the parallel plate method is another high-accuracy method. These methods have a small frequency range, and this shortcoming limits the utility of these methods to analyse broadband textile transmission line substrate dielectric constant variations. Cotton cloth has the largest dielectric constant and loss, according to preliminary data [4].

The wearable antenna can be designed and coated with different materials like alloys, inkjet, and polymer-embedded fabrics, depending on its application and requirement. For instance, textiles are suitable for body wearable antennas built into the clothes. Table 1 illustrates the dielectric properties of normal fabrics.

**Table 1.** The Dielectric Properties of Normal Textile Fabrics.

| Non-Conductive Fabric | $\varepsilon r1$ | tan $\delta$ |
|---|---|---|
| Felt | 1.22 | 0.016 |
| Cordura | 1.90 | 0.0098 |
| Cotton | 1.60 | 0.0400 |
| 100% Polyester | 1.90 | 0.0045 |
| Quartzel® Fabric | 1.95 | 0.0004 |
| Cordura/Lycra | 1.50 | 0.0093 |
| Silk | 1.75 | 0.012 |
| Tween | 1.69 | 0.0084 |
| Panama | 2.12 | 0.05 |
| Jeans | 1.7 | 0.025 |

Fabric antennas require a conducting material to function as a radiator, which is essential for their electrical characteristics [3]; thus, the electric and electromagnetic features of the materials are crucial in antenna design. Alternately, microstrip patch antennas are lightweight and robust, with a low fabrication cost, and are easy to integrate with radio frequency; hence, they are a good candidate for body-worn applications [5].

Flexibility and easy installation in clothes are the main factors of wearable antennas used in the wireless personal area network (WPAN). Furthermore, substrate dielectric constant and its thickness have major impacts on antenna bandwidth and efficiency. To enhance microstrip bandwidth, the substrate thickness for a fixed, relative permittivity is chosen [5]. However, antennas are sensitive to moisture. When an antenna fabric absorbs water, the high dielectric constant of water alters the antenna parameters. Since the textile antennas are in contact with the skin, the probability of the antenna absorbing moisture like sweat is very high. Moreover, there is a risk for antennas to get wet when the clothing is washed. Beyond these effects, tightening the fabric is easier than the other systems [5]. The fabrication technique for each textile material differs. In the textile adhesive technique, the liquid textile adhesive is used to cover the wearable antenna surface. In contrast, the more common sewing technique causes wrinkle formations on the fabric surface. This fabrication technique is also known as computer-aided design (CAD), where a picture is uploaded into software as an embroidery guide for the sewing process.

Table 2 shows the materials used to design wearable antennas. Common materials used in the design of wearable antennas were compared with the most important factors affecting the antenna's performance in terms of the wettability, flexibility, weight and cost to select the best material.

**Table 2.** Antenna Materials Comparison.

| Material | Bending Ability | Robustness to Wetness | Cost | Weight |
|---|---|---|---|---|
| Textile fabric | Medium | Medium | Medium | Low |
| Polymer | High | High | Medium | High |
| Inkjet | Medium | Medium | Low | Low |
| Alloys | High | High | High | Medium |

The designed antenna in [6] achieved a high gain of 6.45 dBi and a Front to Back Ratio (FBR) of 15.8 dB. An e-slot antenna with an electromagnetic band-gap (EBG) and Defected Ground Structure (DGS) printed it into a highly flexible fabric (DGS). On the other hand, fragile conductive inks and/or copper tape were used in [7] state-of-the-art

origami antenna techniques. A copper-equivalent textile-based body-worn antenna [8] was shown to have excellent agreement between simulations and measurements for all e-textile prototypes. E-fibres (metal-coated polymer fibres) and copper were used to make the antenna's specified antenna. In the end, both antennas were analyzed, and it was determined that the cloth antenna performed just as well as its copper counterpart, with achieved gains of around two decibels.

Table 3 compares the antennas from two decades ago with their current predecessors, where the former fared poorly due to some unsuitable materials incorporated in its manufacture. Antenna performance is classified as high, medium or low, based on parameters such as profit and bandwidth (high when the gain > 4; medium when the gain > 2; low when the gain < 2).

**Table 3.** Cloth Antenna History and Its Performance.

| Ref. | Year | Material | Technique | Antenna Performance |
|------|------|----------|-----------|---------------------|
| [9] | 2000 | Erbium | Two shorting strips | High |
| [10] | 2001 | FR-4 | Photonic band-gap structure | Low |
| [11] | 2002 | - | 2 × 2 patch array | Medium |
| [12] | 2003 | Fleece fabric | Fabric substrate | High |
| [13] | 2004 | Felt | Electromagnetic band-gap | High |
| [14] | 2005 | Felt | E-shape patch | High |
| [15] | 2006 | Textile | Bended planer antenna | Medium |
| [16] | 2007 | Textile | Aperture-coupled patch antenna | High |
| [17] | 2008 | Textile | Integrable into protective garments | Medium |
| [18] | 2009 | Electro-textile | MIMO | High |
| [19] | 2010 | Textile | Circuit/full-wave co-optimization techniques | High |
| [20] | 2011 | Fabric | U-slot | High |
| [21] | 2012 | FR-4 | Rectangular loop antenna | Medium |
| [22] | 2013 | FR-4 | Monopole-antenna | Low |
| [23] | 2014 | Textile-fabric | L-shape antenna | Low |
| [24] | 2015 | Duroid 5870 | Coaxial feed line | Low |
| [25] | 2016 | Flexible fabric | Monopole antenna | Medium |
| [26] | 2017 | Flexible Magneto-Dielectric (MD) materials | AMC | Medium |
| [1] | 2018 | Nano-composite | Nano-composite conductive | Low |
| [27] | 2019 | Disk-shaped FR-4 substrate | Circularly polarized button antenna | Low |
| [28] | 2020 | Felt | Meta-material | High |

## 3. Embroidered Textile Antennas

The demand for lighter and more compact personal electronic devices drives the development of wearable electronics and antennas. The incorporation of antennas into everyday clothing would allow for the practicality of smaller, more portable electronic devices without compromising their performance. There are many uses for wearable antennas, including military, aerospace, rescue, medicine, fashion, etc. [29]. Consumers will appreciate the convenience of the system's hands-free operation. Antennas that may be worn as part of clothes that are soft and flexible are often constructed from thin, stretchy conductors [30–32]. For wearable antennas, the trade-off between fabric properties and antenna performance is a major issue.

Connecting personal communications, wireless sensors, and other wireless devices to wireless networks or the "Internet of Things" requires a huge number of antennas. Currently, antennas are integrated into a variety of gadgets, as well as the human body and clothing [30] and this trend will continue in the future. These antennas must be lightweight and bendable. To enable the seamless integration of conventional antennas with clothes, conventional materials like metals and dielectrics must be substituted for conductive and nonconductive fabrics and yarn. Full-textile antennas are antennas constructed entirely from conductive and non-conductive textiles and yarn. Water and water vapour rapidly enter porous conductive and non-conductive fabrics, causing higher losses and dielectric property changes. Changes in input impedance, gain, and radiation patterns result [30].

### 3.1. Types of Conductive Threads

Conductive thread conducts electricity and the textiles are wired. Thomas Edison employed this technology. Carbonized thread satisfies the resistance and lifespan requirements. Conductive threads serve antistatic, electromagnetic shielding, wearables, e-textiles, etc. functions. When sewn onto flexible textiles, they provide a conductive connection like wires [31].

In recent years, there has been a great deal of interest in conductive threads. Typically, textiles are manufactured from insulating materials such as polymers and cottons. Textile threads are made conductive for a variety of purposes; the objective is to achieve EM shielding. Due to their weak conductivity, the surface of insulated textiles accumulates electrical load. Turning the threads conductive makes load transfer and EM shielding possible. Another advantage of conductive threads is their use as wires or connectors in a variety of applications [33–36].

Conductive threads fall into two distinct kinds. There are first intrinsic conductive threads that are composed of conductive fibres. Textiles are composed of conductive materials such as metals (gold, silver, nickel, and steel), or graphite, which can take the shape of wires or threads. They are injected into the structure of textiles or used intrinsically. Each strand is composed of the physically conducting fibre. Metal, graphite, conductive polymers, carbon nanotubes, and other materials are utilised to create these fibres. They have great conductivity but face several obstacles and are heavier and more expensive. Additionally, their structure can damage embroidery equipment because they are less flexible and user-friendly than conventional fabrics. Low-resistance stainless steel is the most popular. These threads are hard to use in a sewing machine as they break while running through needles. Intrinsically conductive polymers include Polyanyline, Polypyrrole, PVA, and PA11. They are thermochemically and environmentally stable. Their remarkable conductivity has drawn notice recently. They are flexible, light, and conduct well, but due to their great cost, they are generally used in research [35–37].

Coated conductive threads are second. Conductive metals like copper, gold, silver, or carbon are implanted or coated onto nonconductive textile cores. Cotton, polymers, or nylon are typical core materials. Various procedures are used to apply these coatings. Coating textiles with metals or conductive materials creates a hybrid. Galvanic coatings are conductive. They face obstacles such substrate suitability, limited adhesion, and corrosion resistance. Metallic salt is another coating technique. This approach reduces conductivity. Coating textiles with metals or other conductive components provides numerous properties; it boosts conductivity and improves structure. Medical, fashion, military, architecture, etc. can use them as they are functional and beautiful. Combining textile core and coating allows stitching. This group's resistance varies by coating material and thickness [31]. Conductive threads are uninsulated and covered with different materials and ply numbers. Not as efficient as a wire, yet it allows current to travel through, removing the need for circuit boards [31].

### 3.2. Challenges in Embroidered Antennas

The challenges of fabricating embroidered antennas are: choose out the best conductive thread for the patch antenna based on factors like conductivity, strength, flexibility, and how the threads behave when stitched together to make an approximately continuous item. Understanding the flow of current on a patch antenna is essential to determine the stitching direction. Increasing antenna efficiency can be achieved by aligning the primary current flow with the stitch direction [32]. This can create further difficulties if the design needs to operate at higher modes when the current is flowing perpendicularly, as well as for more intricate systems where the current is flowing in multiple directions. The effect of different stitching geometry on antenna performance is explored and published in [33]. In general, the tighter the stitching spacing, the greater the efficiency of the antenna. This comes at the cost of decreased flexibility and greater thread length, which immediately results in higher manufacturing costs. The influence of stitching type on the performance of dipole antennas

is described and proven in [38–40] where dipole tag-antennas embroidered with varied thread densities and two different stitch patterns are examined. The authors of [34] describe the embroidery aspects of Ultra High Frequency (UHF) Radio-Frequency Identification (RFID) antenna [32]. Table 4 illustrates some recent research projects in embraided antenna and manufacturing techniques.

**Table 4.** Recent research in embraided antennas.

| Ref | Year | Frequency | Conductive Material | Manufacturing Technique |
|------|------|-----------|---------------------|-------------------------|
| [35] | 2013 | 10.64 MHz | Silver coated nylon yarn | Arudan BEVT-Z1501CB Digital Embroidery Machine |
| [8] | 2014 | 1.9 GHz | Electrically conductive metal-polymer fibres (E-fibres) | Automatic Embroidery |
| [36] | 2015 | 2.4 GHz | e-thread | Embroidery Machine |
| [37] | 2016 | 880–960 MHz/ 1710–1880 MHz | Silverpam yarn | SWF MA-6 Automatic Embroidering Machine. |
| [38] | 2017 | 0.3–3 GHz | 7-filament silver-plated copper Elektrisola E-threads | Automated Embroidery |
| [7] | 2018 | 1015 MHz | 7-filament Elektrisola E-threads | Brother 4500D Embroidery Machine |
| [39] | 2019 | 5 GHz | Conductive weft threads and dielectric warp threads. | commercial embroidery machine |
| [40] | 2019 | 915 MHz | Silver threads | Embroidery Machine |
| [41] | 2020 | 2.4 GHz | Silver | Screen-Printing |
| [42] | 2020 | 6.78 MHz | Silk-coated copper Litz wires | Automated Sewing Machine (PFAFF Creative 3.0) |
| [43] | 2021 | 915 MHz | Silk-coated Litz copper wires | / |
| [44] | 2021 | 868 and 915 MHz | Clevertex silver (brass) hybrid conductive sewing thread | Sewing Machine Bernina QE750 |

## 4. SAR and Antenna Performance with the Human Body

Previous studies have reported the satisfactory performance of wearable antennas in free space, but the same cannot be said when the device is used in proximity with the human body due to frequency shifts and radiation absorption. These drawbacks are contributed by the high theta and decedent dielectric properties of the human body compared to free space [45]. There is a gap in the literature concerning these issues, especially on the effectiveness of dielectric characteristics on the technical performance of wearable antennas.

The on-body performance of the wearable antenna is ranked by quantifying the changes in its input impedance and near-field distribution that are mainly caused by the presence of lossy dielectric material. When the wearable antenna was tested on phantom body fat, there was a 17% increase in impedance, 19% increase in phantom muscle and 20% for the phantom blood. The methods in [45] are recommended in analysing and validating the reason for behaviour degradation when an antenna is operated close to the human body. All these findings are crucial for antenna designers to select the best location for the on-body antenna.

*Impact of EM on Human Body*

Modern electrical appliances are increasing the influence of electromagnetic field (EMF) radiation on humans, and studies reveal the electromagnetic field's impact on human health [46]. In laboratories specializing in the study of antennas, students and researchers are exposed to EMF radiation from the antennas they are studying or manufacturing. On the other side, this absorption that occurs through the human body weakens the performance of the antenna. Fluorescent light emits EM radiation between 380–800 nm, while UV

radiation is 100–400 nm, altering DNA sequence and gene expression. Some people have become electromagnetic radiation sensitive after prolonged exposure to electronic equipment. Some researchers experienced stress, CFS, focus difficulty, allergies, depression, and sleep disturbances. Chronic fatigue syndrome (CFS) is a depressed immune system linked to ELFE. Low- and high-frequency EMF also doubles infection resistance, according to studies.

## 5. Techniques

Researchers have developed techniques to increase antenna performance. Most techniques that have been proven effective are discussed in this paper.

### 5.1. AMC Array Reflector

Artificial magnetic conductors (AMCs) technique can also be referred to as electromagnetic band-gap structures (EBGs) and high impedance surfaces (HISs) that act as a support shield to the wearable antenna. They reduces the waves on the antenna surface resulting in a significant decrease in absorption of radiation by the human body, thus improving antenna gain and front to- back ratio (FBR) [47]. An AMC surface can be a ground plane in some discrete antenna applications [48]. The reflector size is the main component in determining the antenna performance, while an antenna design with multiple layers makes a high-profile antenna system [8].

These structures produce antennas with high gain, low profile, and great efficiency. EBG has sparked interest in the antenna field. EBG structures lower surface wave current, hence enhancing the antenna's performance. Surface waves diminish the antenna's performance [49].

EBG-technique surface wave suppression enhances antenna performance by enhancing antenna gain and antenna efficiency.

It has two interesting characteristics over the frequency range known as a band gap, according to [50] To begin, the reflected wave has the same amplitude as the incident wave. The EBG surface has a similar function to PMC, which does not exist in nature but has a 00-phase reflection. Second, it prevents the propagation of surface waves. EBG has a wide range of applications in antenna design because of its ability to solve typical antenna problems and optimise performance.

### 5.1.1. In-Phase Reflection

The antenna depicted in Figure 2 is a simple wire antenna placed above a PEC ground plane. Under a perfectly conducting electric ground plane, the image current is out of phase with the wire current. In the presence of an EBG or PMC, the subsurface image current would be in phase with the wire current, thereby enhancing the antenna's radiation. EBG operates as an AMC with a reflection phase of +1 (in-phase reflection), as opposed to the conventional metal ground plane, which has a reflection phase of −1 (out-of-phase reflection) (out of phase). The EBG can therefore function as a reflector, capable of redirecting the vast majority of energy in the desired direction.

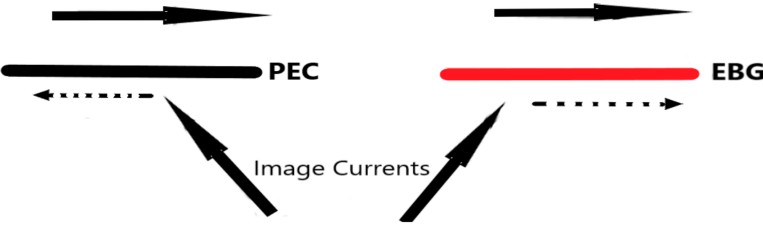

**Figure 2.** Wire antenna with PEC, adapted from [51].

### 5.1.2. Suppressing of Surface Wave

Figure 3 illustrates another characteristic of the EBG surface. It can be used to eliminate the radiation emitted by the ground planes. An "artificial impedance surface" with certain band-gap characteristics inhibits the propagation of surface waves generated by the antenna.

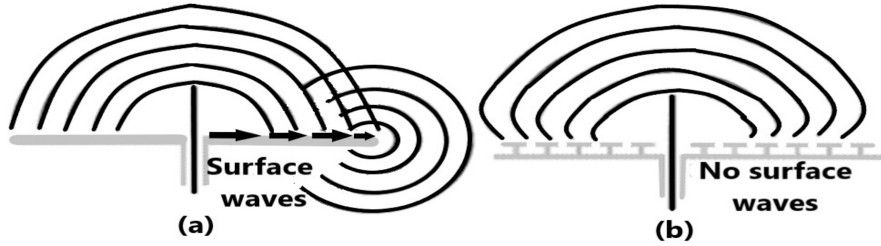

**Figure 3.** The effect of suppressing the surface waves: (**a**) with surface wave, and (**b**) without surface wave, adapted from [51].

The radiation pattern of a monopole antenna on a metallic ground plane is depicted in Figure 4a. Significant characteristics of the antenna pattern include ripples in the forward direction and power loss in the reverse direction.

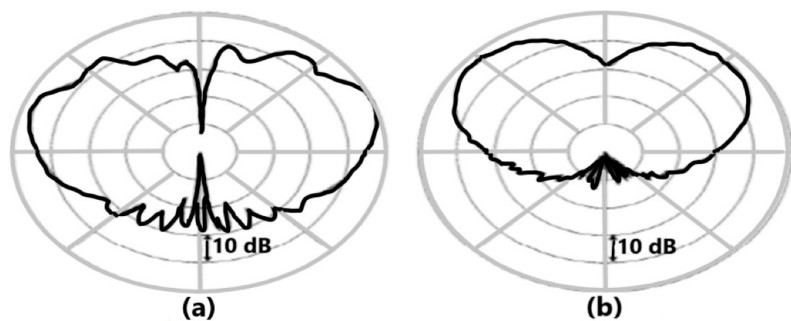

**Figure 4.** Measured radiation pattern of a vertical monopole antenna: (**a**) above PEC ground plane (PEC), and (**b**) above EBG ground plane, adapted from [51].

The surface wave that travels away from the antenna and radiates from the level edges of the ground is responsible for both of these properties. An EBG ground plane is presented in Figure 5b, as a way to reduce the surface wave; as a result, the opposite hemisphere receives less energy and the radiation pattern is more uniform.

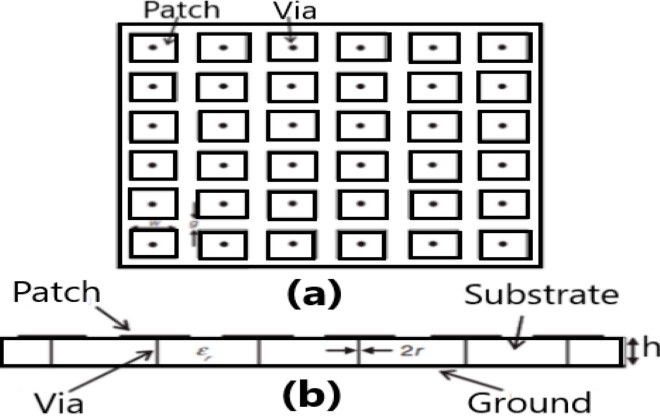

**Figure 5.** EBG surface: (**a**) front view, and (**b**) cross-sectional view, adapted from [51].

In one form, an EBG material is a periodic structure with a high impedance for the propagation of electromagnetic waves within a specific frequency range. In antenna

applications, the antenna can utilise the high impedance or surface wave band gap of an EBG within a specific frequency band. It has been discovered that EBG-structured antennas give significant advantages over conventional antennas [50].

### 5.1.3. EBG Principle

Typically, EBG structures are periodic cells composed of dielectric components or metal. Sievepiper introduced an EBG structure resembling a mushroom for the first time [52]. According to Figure 5, the structure comprises a dielectric substrate, metal patches, a ground plane, and connecting vias.

The operating mechanism of the EBG structure, as depicted in Figure 6, can be explained by an array of LC filters or a parallel resonant circuit. Capacitance is produced by the space between two adjacent patches, whereas inductance is generated by a current loop within the structure through the pin vias. The inductance in an EBG without vias is caused by the ground plane's close proximity to the capacitive array of patches [50].

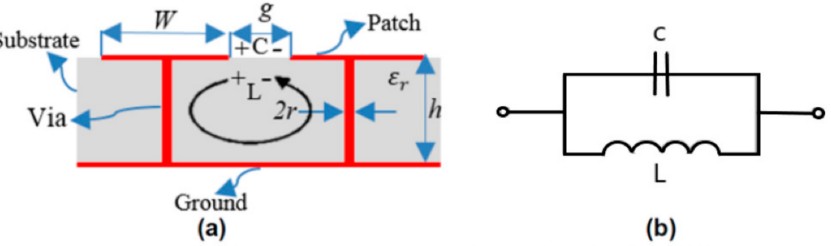

**Figure 6.** EBG unit-cell: (**a**) vias EBG parameters, and (**b**) Lumped Element Equivalent Circuit of the EBG, adapted from [51].

The values of the capacitance (C), inductance (L), bandwidth (BW) and resonant frequency ($f_r$) are given by [50]:

$$L = \mu_0 h \tag{1}$$

$$C = \frac{w\varepsilon_0(1+\varepsilon_r)}{\pi} \cosh^{-1}\left(\frac{w+g}{g}\right) \tag{2}$$

$$f_r = \frac{1}{2\pi\sqrt{LC}} \tag{3}$$

$$BW = \frac{1}{0}\sqrt{\frac{L}{C}} \tag{4}$$

Patch width (W) determines permeability ($\mu_0$), permittivity ($\varepsilon_0$), and impedance ($\eta_0$), all in terms of free space, while g determines the distance between neighbouring patches (g).

Z is the surface impedance at resonance, and it is calculated as follows:

$$Z_s = \frac{j\omega L}{1 - \omega^2 LC} \tag{5}$$

Equations (1)–(5), based on which the parameters of the EBG design were evaluated, [31]. They were the thickness, permittivity, unit cell spacing, and patch width of the substrate. Formula reveals that at low frequencies, the unit cell is typically inductive and so supports the Transverse Magnetic (TM (solid red line)) surface wave. Transverse Electric (TE) waves became dominant as the excitation frequency increased, causing the unit cells to go from resonant to capacitive in a short time. It was found that the surface's impedance was high in a limited frequency region near the resonance frequency. Figure 7 illustrates how the structure inhibited the propagation of surface waves (TE and TM) and directed them toward the frequency band gap (EBG behaviour) [51]. Surface waves were suppressed and more of the system's energy was reflected because of the high level of mismatch [50].

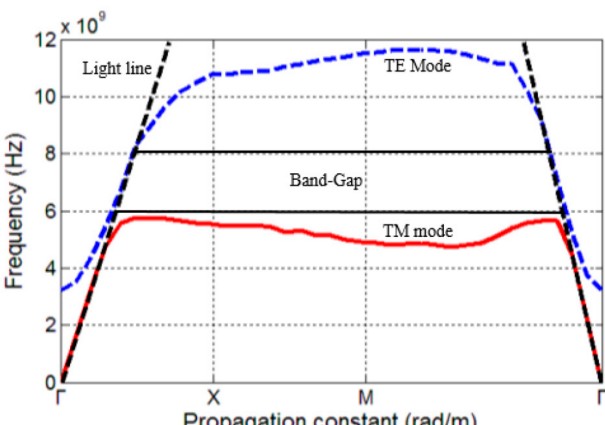

**Figure 7.** Dispersion diagram, adapted from [51].

In [53], a dual-band antenna, 3.5 GHz and 5.8 GHz, backed with a 4 × 4 AMC array, Rogers ULTRALAM 3850 were used for antenna substrate and RO3003 printed as a substrate on AMC array. When measurements were taken at 15 mm from the human body, acceptable gain and SAR were obtained, but after applying a 4 × 4 AMC array over a gap of 1 mm from the human body, the gain was improved by 23.3 dBi and 13.9 dBi, respectively, for both frequencies, while the SAR was reduced by almost 99%. In [48], the designed textile antenna is operated at two frequencies, 2.44 GHz and 5.00 GHz, using an EBG to improve the antenna gain by 3 dB and decreasing backward radiation by 10 dB.

In another study [54], a 92 × 69 mm dual-band antenna showed a stable gain in the passband. The proposed antenna was designed with a small AMC plane in the shape of a square containing a small double square unit integrated with a bow tie for ISM applications. Meanwhile, [55] developed an Artificial Magnetic Conductor (AMC) plane by using different materials (textiles), and both antennas were able to radiate with SAR lower than 2 W/kg (European regulatory standards).

In [56] a wideband monopole antenna array is presenting with uniplanar compact electromagnetic band gap (UC-EBG) structure for high frequency (4.5–6.5 GHz) and high gain of 11.8–13.6 dBi. Moreover, a maximum 1 g SAR value of 0.49 W/kg at 4 GHz and 1 g SAR value of 0.59 W/kg at 6 GHz are achieved when place the antenna at 8 mm away from the human body, which is much lower than FCC standard, guaranteeing the safety for wearable application. The proposed wideband and high-gain wearable antenna array offers some advantages in the wearable application.

### 5.2. Photonic Band-Gap (PBG) Technique

The PBG method has advanced rapidly in the last few years. Resonant cavities, such as the PBG's periodic flaws, can alter the propagation of electromagnetic waves. PBG provides a stopband at a given frequency within the forbidden band gap [50,57].

PBG has been claimed to improve the directivity of antennas, the suppression of surface waves, and the reduction in harmonics.

Because of the periodicity, dielectric contrast, pattern and the repeating spacing between "atoms," PBG is able to stop and reduce the transmission of any electromagnetic wave in a frequency range for space direction [58]. There is no longer a need for additional circuitry in antennas to improve performance, as PBG and PC structures and photonic crystals (PCs) eliminate the requirement for additional weight and size-inducing stop bands by reducing surface wave propagation [59]. Antenna substrates, ground, and covers can all benefit from PBG's ability to reflect radiations in all directions, resulting in increased gain and reduced return loss [60].

PBG Principle

PBG has periodic permittivity. PBG materials are also called photonic crystals due to their closeness to semiconductor physics, where a crystal lattice corresponds to a periodic arrangement of atomic potential [57] (PC). Permittivity periodicity accomplishes for photons what atomic potential does for electrons. Photonic crystal form and index contrast influence many of its optical properties, much as semiconductor conduction qualities. By controlling these two parameters, light may be made to pass through some materials and not others. Scale affects frequency ranges.

Reducing a periodic lattice's elementary cell size raises its frequency spectrum. This allows photonic crystals to be designed for the infrared or visible spectrum. A PBG material for 1–5 GHz has a few millimeters elementary cell size and is easy to build experimentally. The same infrared photonic crystal has 1 m and 0.1 m cell sizes. This frequency range is approximated by swapping the *Y*-axis, which shows energy data, for frequency, or by computing the spectrum, which is a photonic crystal's reflection coefficient. 1D, 2D, and 3D PBG topologies are shown in Figure 8 [57]. This image depicts banned bands. 3D PBG materials are lossless isotropic mirrors for one or more frequency bands. A 2D PBG material behaves as a two-way mirror, as seen in Figure 9. The same substance is transparent at complementary frequencies [57].

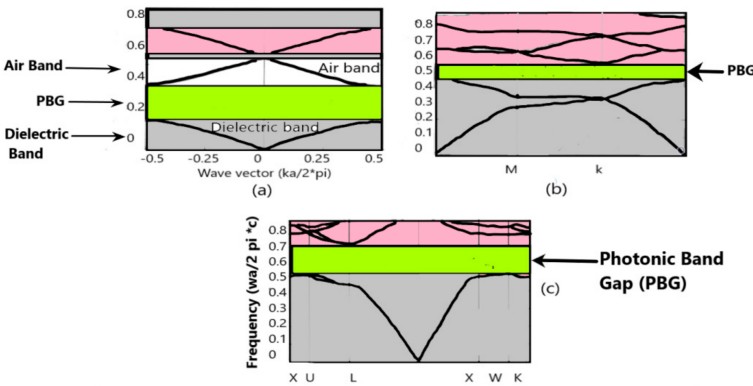

**Figure 8.** (**a**) Crystal dispersion diagram: shaded prohibited bands, lattice period a, light velocity c. This graphic shows forbidden bands. (**b**) Triangular crystal dispersion diagram: shaded restricted bands; air holes in a high permittivity dielectric form the lattice ($\varepsilon = 13$) [57]; (**c**) Crystal dispersion diagram. Shaded restricted bands.

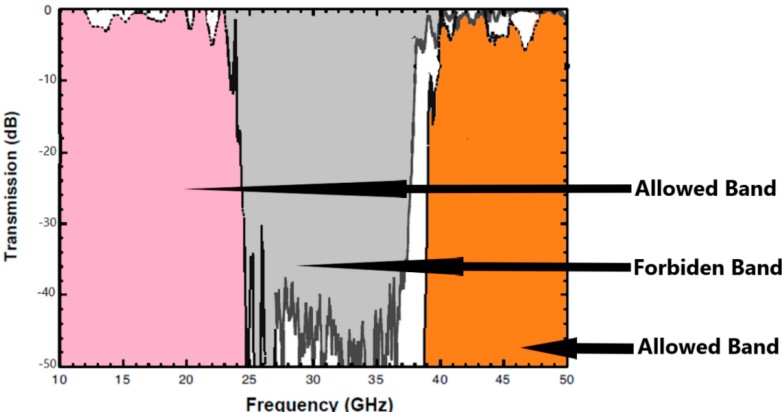

**Figure 9.** Calculated and measured transmission diagram of 18 rows of 11 1.5 mm alumina rods with a 3 mm period (transverse magnetic polarization: electric field E is parallel to rod axis) [57].

The authors of [61] designed a millimeter-wave antenna to improve the antenna patch; PBG substrate and superstrate were created above the radiating element as a cover.

Rectangular-shaped Alumina was utilized as a PBG cover to boost gain (from 7.7 dB to 15.52 dB). The same authors [62] designed PBG planar Inverted-F antenna (PIFA). The PBG structure was placed on the substrate (Teflon $\varepsilon_r$ = 2.65) to decrease surface waves. The design's impedance bandwidth was 27.5% (1.984–2.621 GHz), a 2.1% improvement from ordinary PIFA. The innovative PBG PIFA may be used in wireless 2.4 and Bluetooth systems.

### 5.3. Defected Ground Structure (DGS)

The DGS technique stops the wave propagation through the substrate over a frequency range. It makes slots in different sizes or defects on the ground layer of the microstrip patch antenna (MSP). These mistakes or slots may involve one or more DGS arranged horizontally or vertically on an antenna patch.

DGS Principle

Defects in the ground plane alter the flow of current across the ground plane, leading to various consequences. There are a few parameters that are added to a transmission line (or any other construction) in order to change its properties (slot resistance, slots capacitance, slots and inductance) [50]. Three features differentiate the DGS from the PBG: (1) PBG circuit boards have many periodic structures. A few DGS elements may yield similar properties. DGS shrinks circuits; (2) Both DGS and its circuit are easy to develop and implement; (3) Defect-free structures are more precise. DGS unit and periodic DGS improve its effectiveness. Several unique geometries implanted on the ground plane beneath the microstrip line have been documented [63].

By crafting different-shaped slots inside the patch, this approach reduces antenna size and achieves multi-band [44].

In [45], Duroid (tm) substrate was used to design the proposed antenna and improvement in gain was observed after adding DGS to the antenna and bandwidth and return loss enhancement. On the other hand, a 4 GHz microstrip patch antenna was designed using Glass epoxy FR-4 by, which promoted bandwidth enhancement from 105 MHz to 415 MHz and the gain improvement from 4.12 dB to 6 dB. Meanwhile, the proposed antenna by [59] exhibited a simple improvement in the gain, and it was operated at 2.45 GHz for ISM applications with DGS using different flexible materials like felt (dielectric constant 1.36) and Teflon (dielectric constant 2.1) as a substrate for microstrip antenna. Table 5 compares commonly used techniques highlighted in this paper regarding antenna gain, size of the antenna and influence on fabrication. Table 6 shows how these techniques have been used in recent studies.

**Table 5.** Techniques Comparison.

| Technique | Influence on Gain | Influence on Size | Influence on Fabrication |
|---|---|---|---|
| AMC array | Medium | High | High |
| Photonic band gap | High | High | High |
| Ground structure (DGS) | High | Medium | Medium |

**Table 6.** Recent studies (AMC, PBG, DGS).

| Ref. | Year | Technique | B. W | Gain (dBi) | Substrate | Size (mm) |
|---|---|---|---|---|---|---|
| [64] | 2022 | AMC | 51 | / | FR-4 | 46 × 46 × 1.6 |
| [65] | 2021 | AMC | 5.71 | / | Jeans | 45 × 45 × 2.4 |
| [66] | 2019 | AMC | 14.58 | 6.56 | PDMS | 60 × 60 × 8.5 |
| [67] | 2017 | AMC | / | 5.6 | PDMS | 40 × 60 × 5 |
| [68] | 2018 | AMC | / | 7.5 | Fabric | 60 × 60 × 2.4 |
| [69] | 2016 | AMC | 13.7 | 7.3 | Fabric | 81 × 81 × 4 |

**Table 6.** *Cont.*

| Ref. | Year | Technique | B. W | Gain (dBi) | Substrate | Size (mm) |
|------|------|-----------|------|------------|-----------|-----------|
| [70] | 2018 | AMC | 17 | 6.55 | fabric | 60 × 60 × 2.4 |
| [71] | 2017 | PBG | 21.5 | / | Vacuum | 25 × 25 × 0.8 |
| [72] | 2021 | PBG-DGS | 9.4 | 9.0828/9.2161 | Rogers RT/duroid 5880 (tm) | / |
| [73] | 2020 | PBG | 8.18 | 10 | Rogers 5880 | 44 × 48 |
| [74] | 2019 | PBG | 3.714 | 16.6 | / | 35 × 45 × 3 |
| [75] | 2022 | PBG | / | 1 | / | / |
| [76] | 2022 | DGS | 1.45 | 12.20 | textile | 90 × 100 |
| [77] | 2021 | DGS | 15.7 | 30.3 | / | / |
| [78] | 2021 | EBG-DGS | 9 | 20 | Taconic TLY-5 | 31.1 × 34.7 |
| [79] | 2020 | DGS | 13.5 | / | FR-4 | 58 mm × 58 mm × 1 |
| [80] | 2019 | DGS | 3.82 | 3.5 | FR-4 | 44 × 76 |
| [81] | 2018 | DGS | 2.48 | 65 | FR-4 | 120 × 60 × 1.6 |

## 6. Implemented Wearable Antenna Styles

### 6.1. Helmet and Vest Antenna

Certain professions require specific conditions, tools and equipment to carry out their tasks; for instance, the police force and fire brigade need hands-free wireless devices. The best solution is to design a wearable antenna that can be incorporated into garments such as helmets or vests, but it has become increasingly challenging with the escalation of bandwidth frequency. In addition, it is important to adhere to the radiation safety mandate while maintaining the low-cost and lightweight feature of antennas and fulfil a wide-bandwidth requirement at the same time. The vest and helmet antennas are examples of wide-bandwidth antennas application that matched the needs of their users.

The safety of the wearer is a top priority in antenna design and applications: i.e., in military applications, the fabric antenna is built-in within the helmets. The research in [82] describes how it is better to use air gap to protect users' heads in new military applications instead of energy-absorbing foam. An omnidirectional helmet was introduced by [83] by utilizing the whip antenna with much higher and wider bandwidth, connectivity stability, and flexibility, to be worn on humans than other models. On top of that, the helmet antenna has other benefits like the loss due to the helmet is estimated to be less than 1 dB. Further impedance matching would enhance its gain by 0.2 to 1.2 dB, depending on the operating frequency.

### 6.2. Wearable Monopole and Zip Antenna

A zip antenna is designed to realize a monopole antenna which is widely used in clothes. It is recommended to implement a hidden antenna in a pocket; hence, a pocket zip is preferred for practical considerations. This type of wearable antenna (zip antenna) results in the good performance of Wi-Fi communication systems. The felt material is one of the most commonly used materials to fabricate this type of antennas; for instance, in [3], the textile monopole antenna was designed as a zip antenna operated at 2.5 GHz. A prototype was fabricated and characterised based on the return loss and radiation pattern. It was found that the antenna operated well within 2.4 to 2.7 GHz and suitable for Wi-Fi applications.

### 6.3. Cloth Antenna (Implemented on Clothes)

The usage of the UHF band is preferred in long-range communication devices due to its good propagation characteristics, as such can be found in Industrial Scientific and Medical (ISM) devices. In some special cases, the radio must be hidden under clothes or

implemented on clothes and still maintain the ability to meet the application performance requirements. The antenna design in these applications should be a fabric antenna that can be worn, lightweight and suitable for human body curves to overcome human body effects like radiation and low-profile (below 10 mm) with sufficient bandwidth. For example, [84] implemented a dual-resonance and ultra-wide-band antenna with an operating frequency of 430 MHz with a simple structure and low profile. Figure 10 shows implanted antenna on clothes [85].

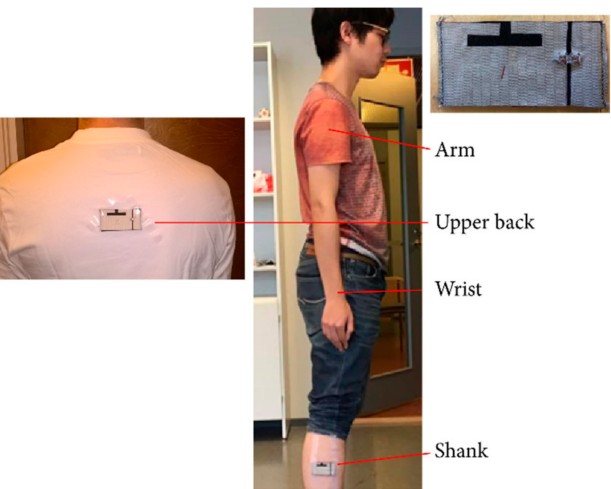

**Figure 10.** Implemented Antenna on Clothes [85].

### 7. Conclusions

Wearable antennas design and applications have been discussed, especially in the medical field. In this paper, the focus was on the factors affecting the performance and efficiency of a wearable antenna to avoid the problems faced by antenna designers. It was concluded that the human body has a strong and fundamental effect on the characteristics and performance of the antenna since a certain percentage of the antenna radiation is absorbed. Therefore, the effect of the human body that was discussed must be taken into account. Based on the effect of external operators on the antenna's behaviour, its manufacture was studied in terms of materials, techniques and shapes that contribute to high efficiency in applications related to human health. Comparisons were made between the common materials and techniques used in manufacturing wearable antennas to maintain their good performance. Based on these comparisons, it is possible to develop antennas with the best specifications and assist the antenna designers in their decision making by considering relevant factors in antenna development. In human activity-related WBNs applications, the future direction for wearable antennas is the production of high-performance fabric substrate antennas that have the capacity to overcome the detrimental effects of the human body.

**Author Contributions:** Conceptualization, methodology and writing—original draft preparation: F.F.H.; Review and supervision: W.N.L.B.M., T.B.A.L. and M.B.O. All authors have read and agreed to the published version of the manuscript.

**Funding:** This research was funded by the Faculty Grant (No: GPF038A-2019) of the University of Malaya.

**Conflicts of Interest:** The authors declare no conflict of interest.

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
