# Peer review of "Key Factors in the Implementation of Wearable Antennas for WBNs and ISM Applications: A Review WBNs and ISM Applications: A Review"

_electronics, doi:10.3390/electronics11152470_

Round 1

Author Response

Response to reviewer 1 comments
Introduction:
It is important to appreciate the reviewer for his/her feedback comments, which helped me improve the quality of my work before anything else.
Secondly, I would like to include in this letter the changes I made in response to the reviewer's comments.

Conceptual issues:
Coherence between different parts:

1-On p.6: “Modern electrical appliances are increasing the influence …” The paragraph does not cohere well with the preceding paragraph. Better to add subsection: impact of EM waves on human body, effects other than radiation-generated heat, etc.

*The paragraph changed to subsection under the subheading "Impact of EM waves on human body " on page 9

2- You speak about the propagation constant but I see an energy band diagram! The mentioned figure is not available in [39] where did it come from?

  • This article's references have been thoroughly checked, and I apologize for citing the incorrect source.
    This error has been fixed, and the correct source has been cited .
  •  
  • 3- On p.13: Where did Figure 9 come from?
  •  
  • Figure 9 also cited to the reference [77].

  • Discussion of results:
    On p.13: Subtracting 5.459 from 5.293 gives 0.166 dB which is merely 3% different between both designs. It is not worth doing it at all!
  • * This search has been removed from the article, as I agree with you completely and there is no need to discuss the results of this research.
    Formatting issues
    Here are some mistake examples that I managed to pick:
    1. On p.1: “detect the body’s body compared to its environment…”
  • * Fixed.
    2. On p.2: Figure 1 takes much space, you can smallen it.
  • *Fixed .
    3. On p.2: The causality between antenna efficiency and permittivity is not clear, you need to rephrase the first paragraph in this paper.
  • * This paragraph has been rewritten, rephrase and the central idea has been elucidated
  • 4. On p.2: Table 1 consider placing in the appendix also for other similar tables.
  • *Fixed .
    5. On p.3: I cannot understand the meaning of this sentence “fabrication technique for each textile material differs, like liquid is different”
  • * This paragraph has been revised to clarify its main idea on page 4.
    6. On p.3: “Error! Reference source not found.” Needs to update reference citation code.
  • * Fixed.
    7. On p.4: Table 2, not so nice how the following appears: “Robust- ness to wetness” try to dimension the table.
  • *Fixed .
    8. On p.4: “[5] indicates a high gain of 6.45 dBi and…” You may consider to open this phrase in a different way.
  • * The sentence rephrased to “The designed antenna in [6] achieved a high gain of 6.45 dBi”
    9. On p.6: I was not able to understand the following sentence: “Students and researchers in labs exposed to EMF radiation, impairing antenna performance.”
  • *The sentence rephrased to be clearer on page 9 .
    10. On p.6: The following acronym has never been identified CFS.
  • * The identify added “Chronic fatigue syndrome (CFS) is a depressed immune system linked to ELFE”
    11. Citation issues in which the reference numbers are not inserted correctly in the text. E.g. p. 6 “It has two interesting characteristics over .. [36,37]”
  • *Fixed .
    12. On p.6: It is confusing to recognize between PBG, EBG, etc. It needs rephrasing.
  • *Rephrasing done.
    13. Citation issues in the figure caption E.g. Figure 3: instead of writing [36,37] there must an indication that you are authorized to reproduce this figure in your work. Same applies for Figure 4 (consider this for all other figures when necessary)
  • *All figures are referenced to their respective sources.
    14. On p. 8: “Sieve paper introduced an EBG …” no reference to this paper.
  • *Reference added .
    15. On p.13: Bad formatting for Figure 8 caption.
  • * Fixed .
    16. On p.13: Style and register error. Kindly use another “DGS trumps PBG. (1) PBG circuit boards have many …” word. Please rephrase.
  • * The sentence rephrased to “Three features differentiate the DGS from the PBG: (1) PBG circuit boards” on page 15.

Reviewer 2 Report

This review manuscript discusses the "Key Factors in The Implementation of Wearable Antenna for WBNs and ISM Applications: A review WBNs and ISM Applications." The manuscript covered most of the previous work. There are a few significant comments.

l  The author seems to follow the related work in "An Overview of Electromagnetic Band-Gap Integrated Wearable Antennas." Can the author emphasize this review article's problem statement and novelty?

l  The wearable cloth antenna was discussed, it would be better to include embroidery-based textile antennas and their challenges too.

l  This paper’s contents come from other review papers, so its novelty is limited. Therefore, I would suggest including something new. Like textile-based embroidered antennas.  Must include the types of conductive threads used. What are the challenges in embroidered antennas? How do integrate these antennas with connectors and other electronic components?

l  The quality of the figures should be improved.

l  In the conclusion, the future directions should be included.

l  As the dielectric constant and tangent loss are important to design the antenna, however, the dielectric constant of cloths is unknown. Thus, it should be determined before starting to design an antenna. I would suggest including some techniques to measure the dielectric parameters of such materials.

l  The authors presented some exciting work at higher frequency bands such as millimeter-wave bands in https://doi.org/10.1038/s41598-020-65622-9 and "10.1109/TAP.2016.2618485". Please try to compare the wearable antenna and EBG work at a higher frequency too.

Author Response

Response to reviewer 2 comments
Introduction:
It is important to appreciate the reviewer for his/her feedback comments, which helped me improve the quality of my work before anything else.
Secondly, I would like to include in this letter the changes I made in response to the reviewer's comments.

l The author seems to follow the related work in "An Overview of Electromagnetic Band-Gap Integrated Wearable Antennas." Can the author emphasize this review article's problem statement and novelty?

* This article aims to serve as a reference for researchers in the subject of wearable antennas, discussing fabric antennas in terms of the materials and techniques used to improve their performance, as well as the issues that these antennas may encounter during their actual implementation.

l The wearable cloth antenna was discussed, it would be better to include embroidery-based textile antennas and their challenges too.

*Embroidery based textile antennas discussed on page 6.

l This paper’s contents come from other review papers, so its novelty is limited. Therefore, I would suggest including something new. Like textile-based embroidered antennas. Must include the types of conductive threads used. What are the challenges in embroidered antennas? How do integrate these antennas with connectors and other electronic components?

• From page 6 to 8 Embroidered antenna discussed deeply in terms of conductive threads types and challenges and table 4 illustrate some recent researches in embraided antenna and manufacturing techniques.

l The quality of the figures should be improved.

• All figures colours improved.

l In the conclusion, the future directions should be included.

*The future directions added to conclusion on page 18.

l As the dielectric constant and tangent loss are important to design the antenna, however, the dielectric constant of cloths is unknown. Thus, it should be determined before starting to design an antenna. I would suggest including some techniques to measure the dielectric parameters of such materials.

• Dielectric Permittivity parameter measurement techniques discussed on page 3 .

l The authors presented some exciting work at higher frequency bands such as millimeter-wave bands in https://doi.org/10.1038/s41598-020-65622-9 and "10.1109/TAP.2016.2618485". Please try to compare the wearable antenna and EBG work at a higher frequency too.

• Ref. [72] added to the article and the proposed design and the results were discussed .

Round 2

Reviewer 1 Report

the document in the updated form is good to go.
Good luck!

Reviewer 2 Report

no further comments